



# The influence of lateral transport on sedimentary alkenone paleoproxy signals

Blanca Ausín[1,2], Negar Haghipour[2,3], Elena Bruni[1], Timothy Eglinton[1]

[1]Geology Department, Salamanca University, Salamanca, 37008, Spain

[2]Earth Sciences Department, ETH Zurich, Zurich, 8092, Switzerland

[3]Laboratory of Ion Beam Physics, ETH Zurich, Zurich, 8092, Switzerland

*Correspondence to*: Blanca Ausín (ausin@usal.es)

**Abstract.** Alkenone signatures preserved in marine sedimentary records are considered one of the most robust paleothermometers available, and are often used as a proxy for paleoproductivity. However, important gaps remain on the

provenance and fate of alkenones, and their impact on derived environmental signals in marine sediments. Here, we analyze the abundance, distribution, and radiocarbon ($^{14}C$) age of alkenones in bulk sediments and corresponding grain-size fractions in surficial sediments from seven continental margin settings in the Pacific and Atlantic Oceans in order to evaluate the impact of organo-mineral associations and hydrodynamic sorting on sedimentary alkenone signals. We find that alkenones preferentially reside within fine-grained mineral fractions of continental margin sediments, with the preponderance of

alkenones residing within the fine silt fraction (2-10 µm), and most strongly influencing alkenone $^{14}C$ age, and SST signals from bulk sediments as a consequence of their proportional abundance and higher degree of OM protection relative to other fractions. Our results demonstrate that selective association of alkenones with mineral surfaces and associated hydrodynamic mineral sorting processes can alter alkenone signals encoded in marine sediments ($^{14}C$ age, content, and distribution) and confound corresponding proxy records (productivity and SST) in the spatial and temporal domain.

## 1. Introduction

Since the initial discovery of alkenones (Boon et al., 1978; Volkman et al., 1980), these molecular biomarkers have become one of the most applied and well-established paleoclimate proxies, allowing estimation of sea surface temperature (SST) and

primary productivity in most oceanographic settings (Sachs et al., 2000; Raja and Rosell-Melé, 2021). Alkenones are long chain ($C_{37}$-$C_{39}$) unsaturated ketones synthesized by some species of haptophytes dwelling in the upper photic zone, most notably the coccolithophore species *Emiliania huxleyi* and *Gephyrocapsa oceanica* (Volkman et al., 1980).

The total abundance of $C_{37}$ alkenones ($C_{37:2} + C_{37:2}$) in marine sediments is widely used as a qualitative proxy for primary productivity on the basis that alkenones are a large component of the total carbon of *Emiliania huxleyi* (Prahl 1988), and that

alkenone degradation is not observed upon zooplankton digestion (Volkman et al., 1980; Grice et al., 1998; Grimalt et al.,





2000). However, this signal can be altered in marine sediments by the significant loss of alkenones that occurs during their export to and deposition on the seafloor. This "flux attenuation" is site-dependent and generally higher during periods of maximum flux (Rosell-Melé and Prahl, 2013). An additional process that may influence this paleoproductivity indicator includes alkenone input via lateral transport of suspended particles and sediments, which has proven to significantly bias the

temperature signal on the Argentine continental margin (Benthien and Müller, 2000) and the Bermuda Rise (Ohkouchi et al., 2002). However, a specific determination of the sediment size fraction in which alkenones may preferentially reside is lacking (Sachs et al., 2000). Given the propensity for preferential mobilization and redistribituion of specific grain sizes (McCave et al., 1995; McCave and Hall, 2006a; Pedrosa-Pàmies et al., 2013; Bao et al., 2016) this information is crucial for assessing potential impacts on sedimentary alkenone signals.

The degree of unsaturation of the $C_{37}$ alkenones, parameterized through the $U^{k'}_{37}$ ratio (Eq. 1), varies as a function of the growth temperature of the precursor organisms.

$$Uk'37 = \frac{C37:2}{C37:2 + C37:3}$$  Eq. (1)

The relationship between $U^{k'}_{37}$ and SST was first quantified in laboratory cultures (Prahl and Wakeham, 1987) with a reported precision of ±0.6°C, leading to the implementation of global calibrations of the $U^{k'}_{37}$ ratio from marine surface

sediments with instrumental SSTs (Müller et al., 1998; Conte et al., 2006; Tierney and Tingley, 2018). The latter calibration curves exhibit larger associated errors because core-top SST does not always effectively record annual average SST from the overlying water column. In regions like the North Atlantic (>48°N), North Pacific (>45°N), Mediterranean Sea, and the Black Sea, systematic $U^{k'}_{37}$-SST decoupling with surface water temperature has been attributed to factors such as seasonal biases in haptophyte productivity and dissolved nutrient concentrations (Epstein et al., 1998), highlighting the need for

seasonally-tuned calibrations (Tierney and Tingley, 2018). Selective degradation of the $C_{37:3}$ due to free radical oxidation and aerobic bacterial processes (Zabeti et al., 2010; Rontani et al., 2013) may result in warmer biases in some settings such as SE Alaska, the eastern Pacific, and Santa Monica Basin (Gong and Hollander, 1999; Prahl et al., 2010; Jaeschke et al., 2017). In other regions, such as the Brazil-Malvinas confluence (Benthien and Müller, 2000; Rühlemann and Butzin, 2006), the Nordic and Labrador Seas (Bendle and Rosell-Melé, 2004; Filippova et al., 2016; Tierney and Tingley, 2018) and

northern Sargasso Sea [*Ohkouchi et al., 2002*], marked SST deviations have been attributed to lateral advection of alkenones synthesized in distal regions characterized by distinct surface ocean temperatures. In this regard, the implementation of a general ocean circulation model indicated that long particle residence times and lateral advection of alkenones (via OM-mineral interaction) could strongly decouple sediment $U^{k'}_{37}$-SST and overlying surface water temperature on continental shelves (Rühlemann and Butzin, 2006). Similarly, advection of pre-aged alkenones associated with mineral surfaces is

typically invoked to explain older radiocarbon ages of alkenones in relation to coeval foraminifera in many continental margin and deep ocean settings (e.g., Mollenhauer et al., 2003; Mollenhauer et al., 2005; Kusch et al., 2010; Ausín et al., 2019).



Sediments deposited on continental margins are the focus of numerous paleoceanographic studies due to the expanded temporal resolution that they offer over deep-sea sedimentary sequences, they thus dominate global calibration data. Yet, an

in-depth investigation of the coupled effects of alkenone-mineral associations and hydrodynamic processes on alkenone-based proxy signals recorded in continental margin sediments has not yet been undertaken. Recent studies have highlighted how the interplay between organo-mineral relationships and the grain-size dependent hydrodynamic mineral particle sorting effects exerts strong control on the content and geochemical signatures of OC in continental margin surface sediments (Bao et al., 2018a; Bröder et al., 2018; Magill et al., 2018; Ausín et al., 2021). In general, fine-grained minerals host higher

amounts of OM than larger particles by virtue of their higher surface area and hence enhanced physical protection against OM remineralization (Keil et al., 1994b; Mayer, 1994a, b; Hedges and Keil, 1995; Keil and Mayer, 2014). Additionally, the size of mineral particles and their propensity for resuspension largely determines their tendency to be remobilized and dispersed at a given bed shear stress (McCave and Hall, 2006a). Consequently, hydrodynamic particle sorting processes not only selectively translocate OC sorbed to minerals but also expose it to further degradation (Bao et al., 2016; Bao et al.,

2018a; Bao et al., 2018b; Ausín et al., 2021). As a component of this OC, alkenones associated with specific grain-size fractions are subject to dispersal and decomposition as a function of the governing hydrodynamic conditions that delineate sediment transport pathways and deposition patterns. Given that the strength and trajectory of mobilizing currents may vary as a function of ocean and climate conditions, and considering continental margins are strategic locations for high-temporal-resolution paleoceanographic investigations, greater understanding of the influence of these mechanisms on alkenone signals

encoded in marine sediments is needed to improve interpretations of derived proxy records. Here, we explore alkenone-mineral grain-size relationships in a suite of surficial sediment samples from seven locations, mostly on productive continental margins, where geochemical evidence exists for the influence of organo-mineral relationships and hydrodynamic particle sorting on OC geochemical signatures and content (Ausín et al., 2021).

**2. Materials and Methods**

**2.1. Surface sediment samples**

Six surface and one near-surface sediment samples were obtained from five different continental margin settings and one

deep-ocean sediment drift (Fig. 1; Table 1). A detailed description of the depositional setting and environmental characteristics of each study site can be found in Ausín et al. (2021). The Peruvian margin site ("PER") is characterized by persistent upwelling that supports very high primary productivity and sustains low oxygen bottom waters (Reimers and Suess, 1983). Sites from Santa Barbara and Santa Monica Basins ("SBB" and "SMB") in the highly productive California margin also feature sub-oxic to anoxic bottom waters favoring OM preservation in underlying sediments. The site

abbreviated as "NAT" is from the New England "Mud Patch", a shelf depocenter south of Cape Cod on the Mid-Atlantic Bight that is characterized by moderately high surface ocean productivity and rapid fine-grained deposition (Twichell et al.,





1981; Goff et al., 2019). The Namibian margin is characterized by strong upwelling and high primary productivity, and the study site "NAM" is under sporadic influence of high-productivity filaments from the adjacent Lüderitz upwelling cell and is located in an OC depocenter on the upper slope produced by the offshore transport of shelf sediments (Inthorn et al., 2006a).

The deep-ocean site "BER" from the Bermuda Rise in the sub-tropical NW Atlantic is characterized by low primary productivity in overlying surface waters and a fully oxygenated water column. This contourite deposit stems from currents associated with deep-ocean recirculation gyres that result in focused deposition of fine-grained sediment (Laine and Hollister, 1981; Laine et al., 1994). The site named as "NAF", on the NW African margin, is influenced by the Canary Current Upwelling system featuring moderate productivity and bottom water oxygen contents (Zonneveld et al., 2010).

Advective sediment transport has been proposed to explain the relatively low settling rates of coccolithophore calcite plates and alkenones (Fischer et al., 2009), in contrast with the minor or negligible presence of pre-aged alkenones (Mollenhauer et al., 2005).

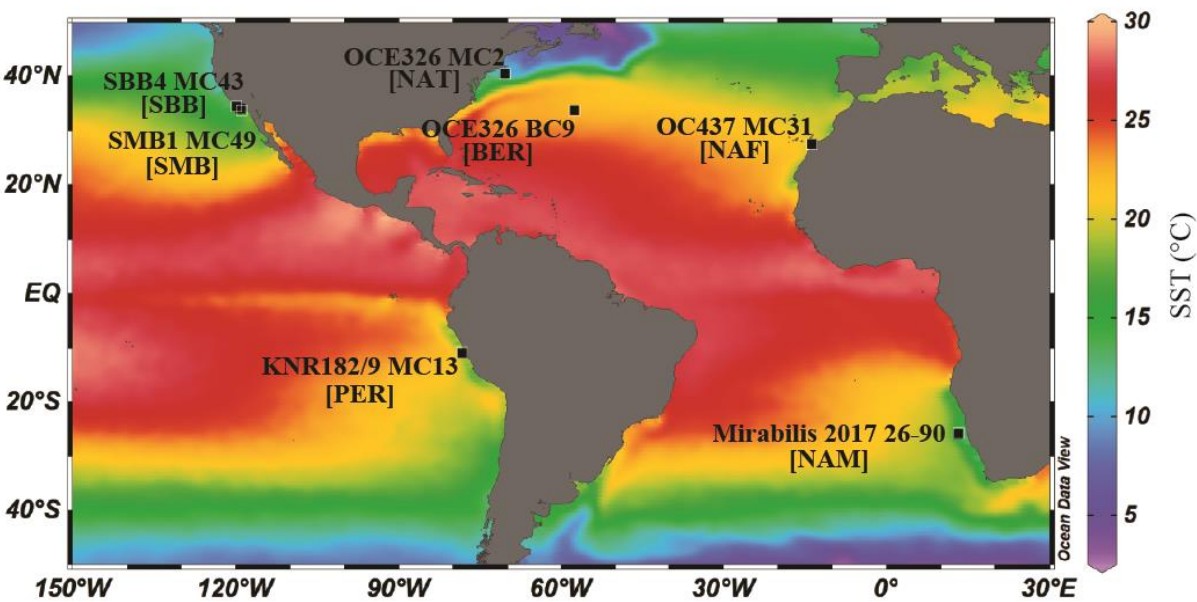

**Figure 1**. Sample location and annual mean SST from the WOA18 product (Locarnini et al., 2019) plotted with the Ocean
Data View (ODV) software (Schlitzer, 2021). Acronyms for each site used in the main text are given within brackets.

Sediment cores were split onboard every 1 cm and stored at -20°C at the Biogeoscience Group ETH Zurich Sample Repository. For each core, samples from the upper 5 cm (Table 1) were freeze-dried, homogenized and fractionated into four grain-size fractions: sand (>300-63 µm); coarse silt (63-10 µm [CS]); fine silt (10-2 µm [FS]); and clay (< 2 µm), as detailed
in Ausín et al. (2021).



**Table 1.** Study sites. Annual mean SST was obtained from the World Ocean Atlas (WOA18) product with a grid of 0.25°longitude by 0.25° latitude (Locarnini et al., 2019) using Ocean Data View (ODV) software (Schlitzer, 2021). MC=multi core and BC=box core. Adopted from Ausin et al. (2021).

| Study site [Acronym] | Sample name | Cruise/year | Longitude | Latitude | Water depth [m] | SST [°C] | NPP [mgC m$^{-2}$ day$^{-1}$] | Depositional setting/Oxygen conditions |
|---|---|---|---|---|---|---|---|---|
| Peruvian margin [PER] | KNR 182/9 MC13 0-3 cm | KNR 182/9 2005 | -78.17 | -11.00 | 326 | 19.42 | 2773 | Outer continental shelf/ Anoxic (OMZ impingement) |
| Santa Barbara Basin [SBB] | SBB4 MC43 0-2 cm | New Horizon 2001 | -119.87 | 34.33 | 340 | 15.28 | 1172 | Lower flank of the basin/ Sub-oxic (OMZ impingement) |
| Santa Monica Basin [SMB] | SMB1 MC49 1-2 cm | New Horizon 2001 | -119.22 | 33.90 | 765 | 16.13 | 1055 | Slightly sloping basin floor/ Anoxic (OMZ impingement) |
| NW Atlantic margin [NAT] | OCE326 MC2 0-3 cm | Bermuda Rise 1998 | -70.54 | 40.46 | 80 | 12.75 | 1276 | Shelf depocenter/ Oxic |
| Namibian margin [NAM] | 2017 26-90 0-3 cm | Mirabilis May 2016 | 13.3 | -26 | 1277 | 16.23 | 1431 | Mid-slope/ Oxic |
| Bermuda Rise [BER] | OCE326 BC9 2-5 cm | Bermuda Rise 1998 | -57.61 | 33.69 | 4517 | 22.51 | 374 | Drift deposit/ Oxic |





| NW African margin [NAF] | OC437 MC31 0-3 cm | Cheeta Cruise 2007 | -13.74 | 27.54 | 1090 | 19.77 | 1377 | Upper continental slope/ Oxic |
|---|---|---|---|---|---|---|---|---|

## 2.2. Alkenone extraction and quantification

An aliquot of 0.5-30 g of dry sediment from bulk and each grain-size fraction was used for total lipid extraction with MeOH/CH$_2$Cl$_2$ (9:1, v/v) using an EDGE® automated extraction system. Resulting total lipid extracts were saponified with 0.5M KOH/MeOH prior liquid-liquid extraction of the neutral fraction with hexane. Silica gel column chromatography was applied to separate the neutral fraction into three fractions of increasing polarity (F$_1$ – F$_3$) using hexane, CH$_2$Cl$_2$, and CH$_2$Cl$_2$/MeOH (1:1 v/v), respectively. F$_2$ fractions, containing the alkenones, were analysed by gas chromatography with flame ionization detection (GC-FID) to determine alkenone C$_{37:2}$ and C$_{37:3}$ concentrations using *n*-hexatriacontane as external standard. Corresponding U$^{k'}_{37}$ ratios were calculated according to equation (1) by Prahl and Wakeham (1987) and transformed to SST values using the calibration of Tierney and Tingley (2018).

## 2.3. Alkenone radiocarbon analyses

The ketone fractions used for determination of alkenone concentration and unsaturation were further purified for compound specific [14]C analysis following Ohkouchi et al. (2005), with purity of isolated alkenone fractions assessed via GC-FID. Purified samples were subsequently transferred into tin elemental analyzer (EA) capsules with CH$_2$Cl$_2$ (3 ×50μL). The solvent was removed on a hot plate at 35°C prior wrapping the samples. Blanks were prepared in the same fashion as the samples and spiked with varying masses of oxalic acid II (OXAII; modern [14]C age; Δ[14]C 1.34 ‰) and phthalic anhydride (PHA; infinite [14]C age; Δ[14]C 0 ‰) reference standards in order to quantify and characterize contamination introduced during sample preparation. Samples, spiked blanks, and solvent and capsule blanks were measured within 20 h of preparation as CO$_2$ using an EA system interface coupled to a gas ion source (GIS)-equipped Minicarbon Dating System (MICADAS) (Synal et al., 2007; McIntyre et al., 2016) at the Laboratory of Ion Beam Physics, ETH Zürich. Data assessment was performed with the BATS data reduction software (Wacker et al., 2010). The model by Hanke et al. (2017) was applied to correct for constant contamination. The estimated mass of extraneous carbon was 1.12 ±0.22 μg C with F[14]C value of 0.99 ±0.2.

## 3. Results

### 3.1. Alkenone concentration and distribution





The fraction-weighted alkenone concentration is comparable to bulk values in PER, NAF, NAM, BER and SMB samples (Table 2), implying a 100-88% alkenone recovery. The large discrepancy between bulk and fraction-weighted alkenone concentrations in SBB and NAT suggests significant loss of alkenones occurred during sediment fractionation in SBB (fraction-weighted values < bulk values) and during manual column chromatography of bulk sediments in NAT (fraction-weighted values > bulk values). Alkenone concentrations in bulk sediments are highest in PER (17898 ng gdw$^{-1}$), and

decrease in the order NAM > SMB > SBB > NAT > NAF, with minimum values in BER (28 ng gdw$^{-1}$; Fig. 2A and Table 2). With the exception of NAF and BER sediments, where the clay fraction hosts the largest proportion of alkenones followed by FS, alkenone concentrations are highest within the FS fraction at all sites. Alkenone concentrations normalized to OC$_\%$ also show that the OC in the smallest grain sizes (FS and Clay) are associated with the highest alkenone abundances (Fig. 2B).

**Table 2.** Alkenone concentration and derived SST values. TOC and fractional abundance of grain-size fractions from bulk sediments (Bulk$_\%$) are taken from Ausin et al. (2021)

| Site | Sample | $C_{37:3}$ [ng/g] | $C_{37:2}$ [ng/g] | Alkenone concentration [ng/gdw] | Abundance weighted average concentration [ng/gdw] | Alkenones normalized to TOC [ng/g OC] | $U^{k'}_{37}$ | $U^{k'}_{37}$-SST [°C] | Abundance weighted average SST [°C] | TOC [wt%] | Bulk$_\%$ [%] |
|---|---|---|---|---|---|---|---|---|---|---|---|
| PER | Bulk | 3985 | 13914 | 17898 | 17905 | 108642 | 0.78 | 21.72 | 22.05 | 13.19 | |
| | Sand | 617 | 2649 | 3266 | | 43259 | 0.81 | 22.70 | | 7.55 | 0.2 |
| | CS | 2684 | 9163 | 11847 | | 130325 | 0.77 | 21.60 | | 9.09 | 38.6 |
| | FS | 5551 | 21963 | 27514 | | 157045 | 0.80 | 22.33 | | 17.52 | 48.4 |
| | C | 31 | 122 | 153 | | 2940 | 0.80 | 22.36 | | 5.21 | 12.8 |
| SBB | Bulk | 877 | 1028 | 1905 | 1905 | 81391 | 0.54 | 14.72 | 13.85 | 2.34 | |
| | Sand | 618 | 770 | 1389 | | 41823 | 0.55 | 15.17 | | 3.32 | 2.6 |
| | CS | 112 | 110 | 222 | | 20391 | 0.50 | 13.41 | | 1.09 | 63.6 |
| | FS | 1038 | 1269 | 2306 | | 66462 | 0.55 | 15.03 | | 3.47 | 27.2 |
| | C | 762 | 676 | 1438 | | 75699 | 0.47 | 12.68 | | 1.90 | 6.7 |

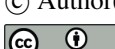


| | | | | | | | | | | | |
|---|---|---|---|---|---|---|---|---|---|---|---|
| SMB | Bulk | 1245 | 1654 | 2899 | 2899 | 129671 | 0.57 | 15.64 | 15.18 | 2.20 | |
| | Sand | 2014 | 2823 | 4837 | | 188023 | 0.58 | 16.02 | | 2.55 | 2.7 |
| | CS | 54 | 63 | 117 | | 19884 | 0.54 | 14.61 | | 1.08 | 56.9 |
| | FS | 2860 | 4096 | 6956 | | 224523 | 0.59 | 16.17 | | 3.00 | 32.0 |
| | C | 916 | 1115 | 2031 | | 266302 | 0.55 | 15.00 | | 0.77 | 8.3 |
| NAT | Bulk | 347 | 467 | 814 | 2548 | 72140 | 0.57 | 15.74 | 14.30 | 1.03 | |
| | Sand | 45 | 96 | 141 | | 39820 | 0.68 | 18.90 | | 0.33 | 11.2 |
| | CS | 87 | 95 | 182 | | n/d | 0.52 | 14.17 | | 0.65 | 47.6 |
| | FS | 3792 | 3583 | 7375 | | 287564 | 0.49 | 13.14 | | 2.43 | 31.0 |
| | C | 793 | 775 | 1568 | | 163044 | 0.49 | 13.38 | | 2.47 | 10.3 |
| NAM | Bulk | | | 7649 | 8217 | 67612 | | | 16.62 | 6.27 | |
| | | 2921 | 4728 | | | | 0.62 | 17.03 | | | |
| | Sand | 4228 | 7735 | 11963 | | 313734 | 0.65 | 17.87 | | 5.41 | 10.7 |
| | CS | 2547 | 3826 | 6373 | | 157689 | 0.60 | 16.51 | | 5.85 | 68.5 |
| | FS | 6122 | 9156 | 15278 | | 317896 | 0.60 | 16.48 | | 6.98 | 16.8 |
| | C | 57 | 78 | 135 | | 1869 | 0.58 | 15.80 | | 4.03 | 4.0 |
| BER | Bulk | 11 | 17 | 28 | 26 | 6751 | 0.60 | 16.37 | 20.23 | 0.42 | |
| | Sand | n/d | n/d | n/d | | n/d | n/d | n/d | | 0.32 | 5.5 |
| | CS | 3 | 6 | 9 | | 7340 | 0.69 | 19.02 | | 0.12 | 49.2 |
| | FS | 9 | 13 | 22 | | 4582 | 0.59 | 16.15 | | 0.47 | 33.4 |
| | C | 10 | 113 | 123 | | 0 | 0.92 | 25.77 | | 0.50 | 11.9 |
| NAF | Bulk | 197 | 537 | 734 | 723 | 48976 | 0.73 | 20.38 | 20.45 | 0.85 | |
| | Sand | 272 | 278 | 549 | | 49219 | 0.51 | 13.72 | | 0.63 | 4.9 |
| | CS | 143 | 421 | 564 | | 59259 | 0.75 | 20.80 | | 0.53 | 49.1 |
| | FS | 174 | 510 | 684 | | 29389 | 0.75 | 20.78 | | 1.30 | 35.6 |
| | C | 421 | 1256 | 1677 | | 69135 | 0.75 | 20.88 | | 1.39 | 10.5 |

The relative proportion of the di- and tri- unsaturated alkenones exhibit significant variability among grain size fractions at each site (Figs. 2C and D). With the exception of NAF, the proportion of alkenone $C_{37:3}$ is lower in the sand fraction in relation to the bulk in all samples, while no other clear distributional pattern is observed among size classes.





**Figure 2.** Alkenone concentration in bulk sediment and size fractions. A) Total $C_{37}$ alkenone amount per gram of sediment, B) Total $C_{37}$ alkenone amount normalized to $OC_\%$, C) $C_{37:2}$ alkenone relative amount, and D) $C_{37:3}$ alkenone relative amount.

**3.2. Alkenone radiocarbon ages**

Alkenone [14]C ages in bulk sediments were measured in four samples (PER, SMB, NAT, and NAM) (Fig. 3 and Table 3). Alkenone ages vary among sites, ranging from 2300 [14]C yr in NAM to 500 [14]C yr in PER. Comparison of these results with bulk OC and planktic foraminifera [14]C ages from the same samples (Ausín et al., 2021) shows alkenones and OC ages are 175 comparable, and both are older than corresponding planktic foraminifera [14]C ages.





**Table 3.** Alkenone radiocarbon analysis. Measured mass, raw and corrected fraction modern (F$^{14}$C), corrected radiocarbon ages and corresponding 1σ errors. Radiocarbon ages and associated 1σ uncertainties have been rounded according to convention.

| Lab code | Site | Sample | Mass (ug C) | Raw F$^{14}$C±1σ | Corrected F$^{14}$C±1σ | Radiocarbon age ($^{14}$C yr BP) ±1σ |
|---|---|---|---|---|---|---|
| ETH- | PER | Bulk | 46 | 0.9402 ±0.0070 | 0.9402 ±0.0089 | 490 ±75 |
| 95663.1.1 | PER | CS | 78 | 0.9300 ±0.0169 | 0.9300 ±0.0074 | 580 ±65 |
| 95664.1.1 | PER | FS | 59 | 0.9473 ±0.0045 | 0.9473 ±0.0080 | 4310 ±70 |
| 95665.1.1 | SMB | Bulk | 56 | 0.9271 ±0.0048 | 0.9271 ±0.0091 | 610 ±80 |
| 95666.1.1 | SMB | Sand | 68 | 0.7345 ±0.0065 | 0.7345 ±0.0068 | 2480 ±75 |
| 95667.1.1 | SMB | FS | 86 | 0.8618 ±0.0029 | 0.8618 ±0.0074 | 1190 ±70 |
| 95668.1.1 | SMB | C | 27 | 0.6830 ±0.0033 | 0.6830 ±0.0094 | 3060 ±110 |
| 95669.1.1 | NAT | Bulk | 51 | 0.8623 ±0.0034 | 0.8623 ±0.0080 | 1190 ±75 |
| 95670.1.1 | NAT | Sand | 30 | 0.6812 ±0.0025 | 0.6812 ±0.0123 | 3080 ±150 |
| 95671.1.1 | NAT | FS | 91 | 0.8344 ±0.0025 | 0.8344 ±0.0077 | 1450 ±75 |
| 95672.1.1 | NAM | Bulk | 66 | 0.7436 ±0.0023 | 0.7436 ±0.0069 | 2380 ±75 |
| 95673.1.1 | NAM | Sand | 47 | 0.6993 ±0.0021 | 0.6993 ±0.0066 | 2870 ±75 |
| 95674.1.1 | NAM | CS | 48 | 0.7692 ±0.0023 | 0.7691 ±0.0069 | 2110 ±70 |
| 95675.1.1 | NAM | FS | 107 | 0.7108 ±0.0050 | 0.7108 ±0.0060 | 2740 ±70 |


Purification of alkenones for radiocarbon dating was possible in some of the size fractions for these four samples. Alkenones contained in sand fractions are the oldest, while those hosted within FS and CS show the smallest age offsets with respect to bulk sediments (Fig. 4).

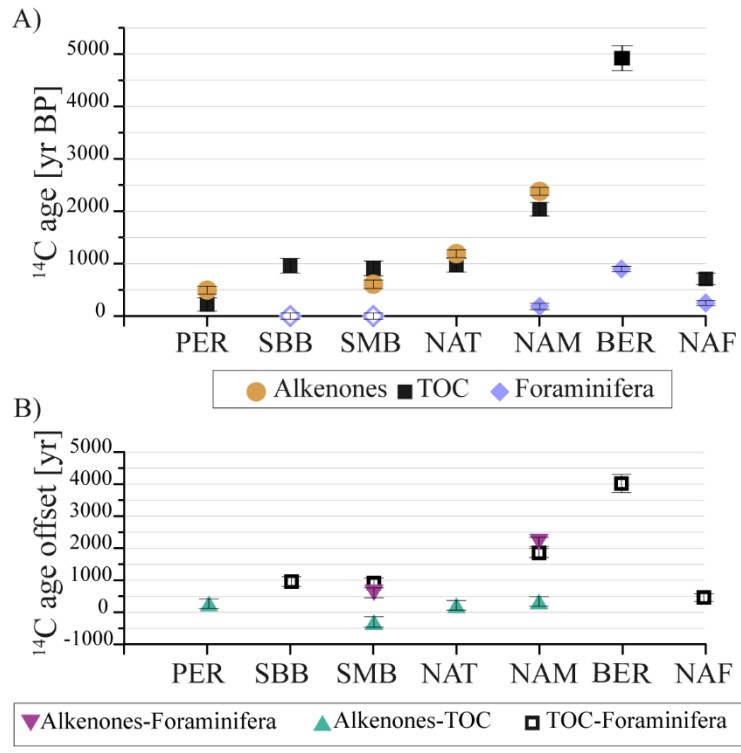


**Figure 3.** Radiocarbon ages of alkenones (this study), TOC and planktic foraminifera (Ausín et al., 2021) from bulk sediment samples (A) and age discrepancies among them (B). Open diamonds indicate foraminifera that incorporate bomb $^{14}$C.





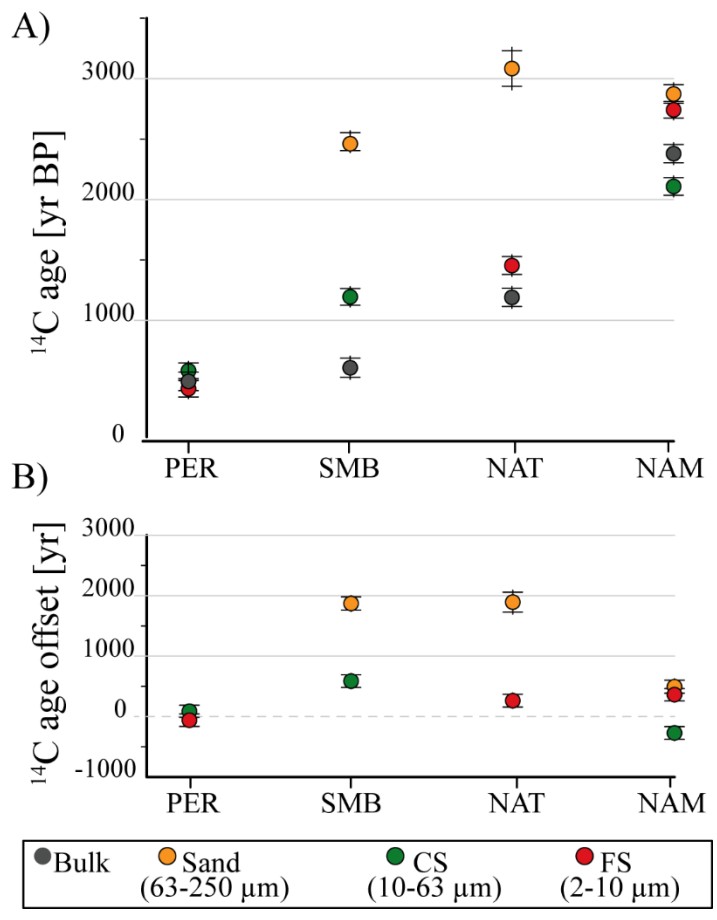


**Figure 4.** Radiocarbon ages of alkenones contained in bulk and grain-size fractions at each study site (A) and age discrepancies between $^{14}$C ages of alkenones in size fractions and corresponding bulk sediment.

## 3.3. Alkenone-SST

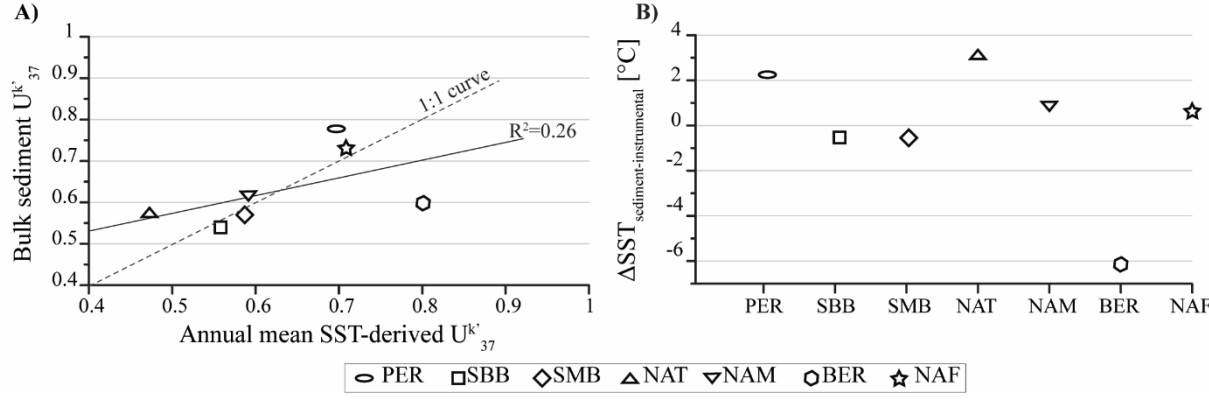






**Figure 5.** SST from bulk sediments and atlas data (Locarnini et al., 2019). A) $U^{k'}_{37}$ ratio from bulk core-top sediments compared to $U^{k'}_{37}$ ratio calculated from atlas annual mean SST (Locarnini et al., 2019). B) Comparison of SST from bulk sediments and atlas annual mean SST.

$U^{k'}_{37}$ ratios and corresponding alkenone-SST values from bulk sediments show a weak positive relationship with annual-mean SST observations ($R^2$=0.26) (Figs. 5A). Only SST from samples from SBB and SMB are comparable to atlas data, whereas temperature differences ranging from -6°C to 3°C are observed at the other sites (Fig. 5B). Abundance-weighted average SST of the analyzed grain-size fractions compares relatively well with bulk SST except at BER (Table 2), which shows a -4.4°C difference. The latter is attributed to the lack of detectable alkenones in the sand fraction of BER. Except for

BER, SST discrepancies imply core-top SST is warmer than surface water temperature. $U^{k'}_{37}$-SST shows significant variability among size grain size fractions at each site (Fig. 6A). The smallest SST variation among size fractions is observed at PER, SMB and NAM. Sand shows the warmest temperature signal in relation to other fractions at 5 out of 6 locations (Fig. 6). Overall, FS shows the smallest temperature offsets with bulk sediment (Fig. 6B). No specific fraction shows larger/smaller offsets with annual averaged SST (Fig. 6C).

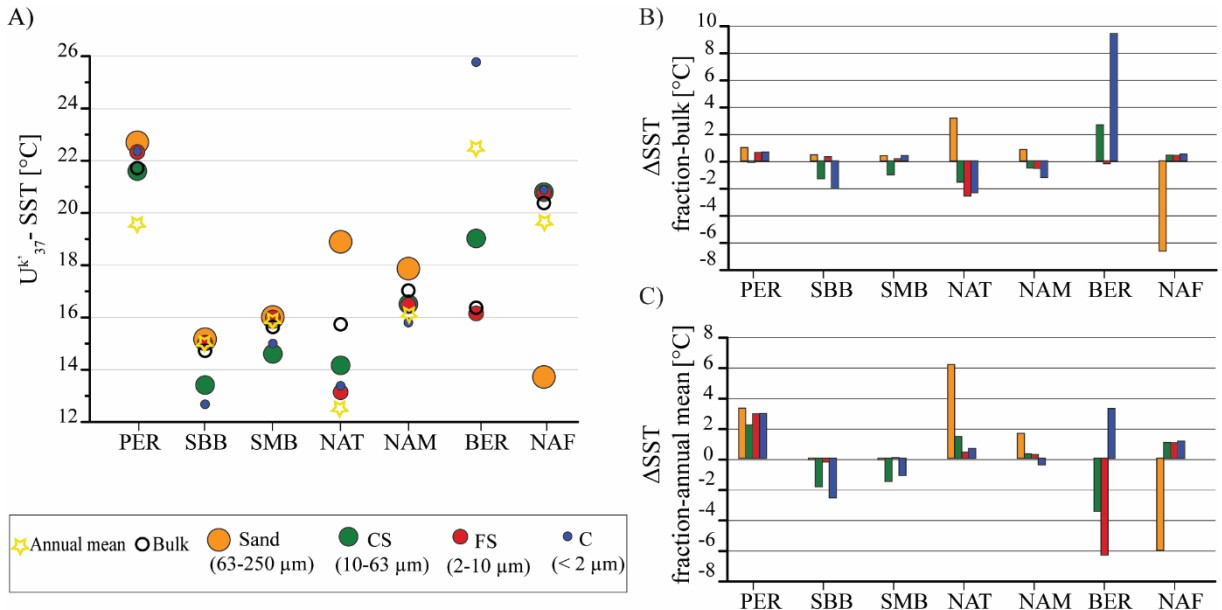


**Figure 6**. SST at each site A) Bulk-, grain-size, and annual mean SST (Locarnini et al., 2019). Temperature difference between B) bulk- and annual mean SST and C) each grain-size fraction and bulk- SST.

**4. Discussion**





### 4.1. Alkenone signals (and biases) from bulk sediment samples

Alkenone concentrations in bulk sediments follow the identical pattern to that of $OC_\%$ ($R^2$=0.99, n=7), indicating similar
preservation mechanisms for both and that bulk OC is predominantly derived from marine primary production at each
location (Fig. 7A). These results support the hypothesis that alkenone fate in marine sediments is largely influenced by
organo-mineral relationships and hydrodynamic mechanisms (Ausín et al., 2021).

Older-than-foraminifera alkenone ages indicate contributions of pre-aged alkenones in the four samples analyzed (Fig. 3),
and previously observed at the three other studied regions: Santa Barbara Basin, Bermuda Rise, and, to a lesser extent, NW
African margin (Ohkouchi et al., 2002; Mollenhauer and Eglinton, 2007). These results imply that alkenone signatures are
influenced by processes such as bioturbation, preferential degradation of fresh alkenones, and/or translocation of older
alkenones (e.g., lateral advection via entrainment in sediment resuspension-deposition cycles or nepheloid layers) associated
with along- or across-margin transport. A significant influence from bioturbation is unlikely since all sites are characterized
by high sedimentation rates (>20 cm/kyr) (Bothner et al., 1981; Wefer et al., 1990; Schaaf and Thurow, 1995; Ohkouchi et
al., 2002; Mollenhauer et al., 2005; Inthorn et al., 2006b; Balestra et al., 2018), and given that some sites (e.g., SMB) contain
varved sediments deposited under the influence of anoxic or sub-oxic bottom waters. In contrast, prolonged particle aging
due to resuspension and downslope transport is a feature of OC-rich continental margin sediments (Mollenhauer et al.,
2008). The joint assessment of $^{14}$C ages and SSTs among grain-size sediment fractions at each site provides insights into the
influence of selective degradation and alkenone translocation mechanisms (Secs. 4.3. and 4.4.).

Older alkenones may carry a different temperature signal than that of the water column overlying the depositional site if they
originate from a distal location or were synthesized during colder/warmer past periods. Alkenones from SMB and SBB are
found to accurately reflect local instrumental SST (Fig. 5), while a +1-3°C discrepancy (towards warmer temperatures) is
observed at other locations with the exception of BER (-6 °C). In both cases, these temperature discrepancies exceed the
analytical uncertainty. Such a warmer bias is a common feature of sediments from many locations with the exception of
those underlying tropical waters (Conte et al., 2006; Prahl et al., 2010). Previous authors argue that this bias cannot be solely
explained by faster degradation of the more-unsaturated $C_{37:3}$ alkenone (Rosell-Melé et al., 1995) and ascribe it to seasonal
production and/or lateral transport of alkenones (Goñi et al., 2001; Sachs and Anderson, 2003; Conte et al., 2006). With
respect to BER, we used sub-surface sediments (2-5 cm; foraminifera $^{14}$C age=900±50 yr), but alkenone-SST from the core-
top (0-1 cm) of the exact same core (Ohkouchi et al., 2002) leads to a -6.6°C (cold) bias. Recent evidence on the advection
of lithogenic particles from the shelf of the NE Canadian maritime provinces (Nova Scotia, Newfoundland) supports lateral
transport of alkenones from these colder and more productive waters, previously proposed to explain hydrogen isotope and
$^{14}$C-depleted values of alkenones at this site (Ohkouchi et al., 2002; Englebrecht and Sachs, 2005; Hwang et al., 2021), and
consistent with the cold bias found at BER.

In light of the strong agreement between $OC_\%$ and alkenone concentration and the temporal and temperature biases observed
in bulk surface sediments from all the studied sites, we speculate that alkenone-proxy signals from continental margin





sediments can be strongly modulated by the interplay between organo-mineral relationships and differential hydrodynamic sorting of mineral particle sizes. Alkenone concentrations, $^{14}$C ages and U$^{k'}_{37}$-SST values measured on specific grain-size fractions provide a means to evaluate this hypothesis.

**4.2. Influence of hydrodynamic sorting processes on sedimentary alkenone signals**

Despite evidence of substantial alkenone loss during sample workup in one or several size fractions from SBB and bulk sediments from NAT, the overall strong positive correlation between alkenone concentration and OC$_{\%}$ in sediment fractions ($R^2$=0.81) indicates mutual preservation mechanisms also exist within mineral grain size classes (Fig. 7A). Hence, and as

observed for OC (Ausín et al., 2021), the large differences in alkenone concentration among grain size fractions correspond to preferential association with, and protection by mineral grains having greater surface area (i.e., FS) (Premuzic et al., 1982; Keil et al., 1994a; Keil et al., 1994b) and to further exposure to degradation for alkenones (and associated organic matter) residing in the least-cohesive grain size fraction that is more prone to resuspension (i.e., CS) (McCave et al., 1995; McCave and Hall, 2006b).

When alkenone concentrations in size fractions are normalized to the bulk sediment mass, the primary contribution of FS — and CS to a lesser extent — is apparent (Fig. 7B). Given the propensity of FS to resuspension and mobilization under strong currents (McCave and Hall, 2006b), we argue that the temporal offsets and temperature biases observed in bulk sediments can be largely ascribed to the lateral supply of pre-aged/allochthonous alkenones sorbed onto the surfaces of fine-grained, mobilizable (fine-silt) minerals. To a lesser extent, advection of coarser grains (i.e., CS) can also contribute significantly to

signals embedded in bulk sediments. Consistent with this notion, FS shows the smallest age and temperature offset with respect to corresponding bulk sediments (Fig. 4B and 6B). In addition to SST and temporal offsets, our results suggest that the alkenone-based productivity proxy (Raja and Rosell-Melé, 2021) may also be influenced by the translocation and deposition of fine sediments from distal regions. Its impact can be particularly relevant in regions where the contribution of silt minerals to the bulk sediment mass is significant.


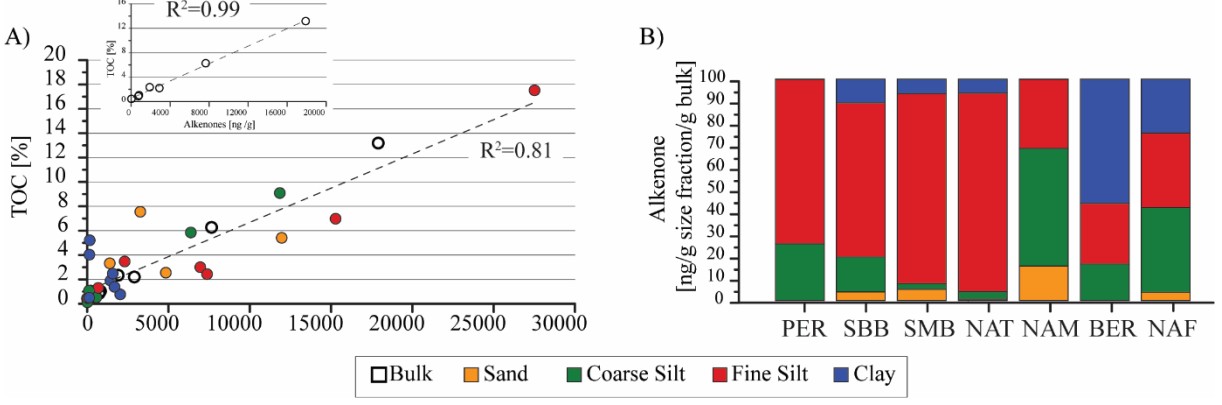





**Figure 7**. Alkenone correlation with $OC_{\%}$ from Ausín et al. (2021) in bulk sediments (upper left panel) and grain size fractions (A) and within each size fraction per gram of bulk sediment (B).

**4.3. Selective alkenone degradation during lateral particle transport**

The strong grain-size dependence of OC-$^{14}$C ages found in all the study sites (Ausín et al., 2021) is not uniformly observed for the more limited alkenone-$^{14}$C age data set (Fig. 4). Yet, the strong positive linear relationship observed between both ($R^2$=0.78) suggests alkenones could exhibit a similar age-grain-size relationship driven by the differential influence of hydrodynamic processes on mineral grain sizes (Fig. 8). Overall, FS and sand show warmer-than-instrumental alkenone-derived SSTs at most sites (Fig. 6C), with the sand fraction exhibiting the greatest warm bias and oldest ages. These results may reflect extensive diagenetic alteration as a consequence of two non-exclusive mechanisms: i) input of pre-aged alkenones synthesized in warmer waters, or ii) selective microbial or abiotic oxidative degradation of more labile (fresher) polyunsaturated ($C_{37:3}$) alkenones as a consequence of decreasing mineral protection with increasing grain size. Although there has been evidence for (Gong and Hollander, 1997; Hoefs et al., 1998) and against (Sikes et al., 1991; Teece et al., 1998; Grimalt et al., 2000) the impact of selective alkenone degradation on sediment $U^{k'}_{37}$ ratios, more recent work has demonstrated that autoxidation and aerobic bacterial degradation can cause selective degradation of more unsaturated alkenones, altering corresponding $U^{k'}_{37}$ ratios, resulting in warm temperature biases of up to 5.9 °C (Rontani et al., 2013, and references therein). Given that the relative increase of SST and $^{14}$C age is most pronounced for the [comparatively immobile] sand fraction, and that it is difficult to envision how advected alkenones systematically carry a warmer signal than that of the overlying water column for diverse locations, we suggest that selective degradation of $C_{37:3}$ provides the most viable explanation. While further evidence is required in order to attribute warmer biases to selective degradation of $C_{37:3}$ within specific size fractions, a universal SST-grain size relationship is not expected because corresponding SST depends on the temperature of surface waters where alkenones were produced. In this context, a much colder initial surface ocean signal than that at the depositional location could mask the influence of selective degradation during oxic transport.





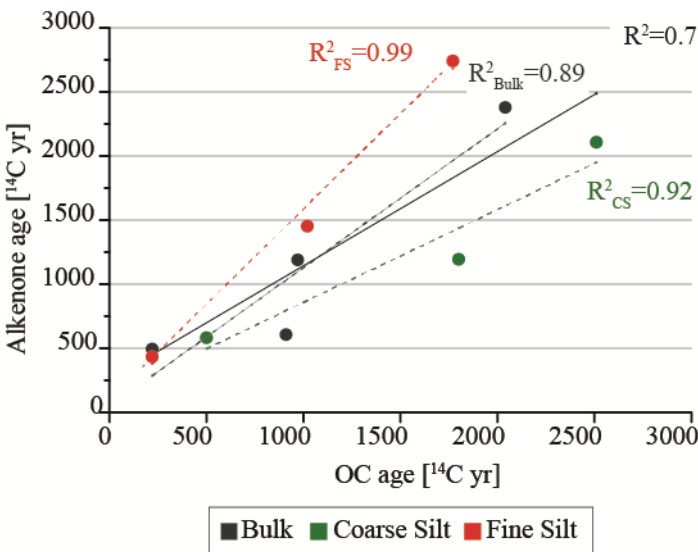

**Figure 8**. Relationship between OC- (Ausín et al., 2021) and alkenone- $^{14}$C ages. The black solid regression line considers all the data (n=10; $R^2$=0.71).

## 4.4. Site-specific hydrodynamic mechanisms

Alkenone ages from all PER grain size fractions are similar and close to modern values (Fig. 4). This observation, along with the largest alkenone concentrations, are attributed to the high vertical flux of fresh OM (Reimers and Suess, 1983) and agree with a minor, although discernable, effect of hydrodynamic sorting on OC signals (Ausín et al., 2021). Alkenone-derived SST values of size fractions and bulk sediments at PER are similar, but differ markedly from (are 2.3°C warmer than) instrumental SST (Fig. 6) (Prahl et al., 2010; Kienast et al., 2012). Resuspension of recently deposited bottom sediments from the shelf and offshore transport as suggested by Pak et al. (1980) to explain particle advection maxima at 140-200 m water depth at this location is discarded because across-shelf transport would hypothetically translocate a colder signal. Selective degradation of $C_{37:3}$ seems unlikely, as this is expected to exert a differential impact on the $U^{k'}_{37}$ of the different fractions based on their size (i.e., propensity for resuspension and OM exposure to oxic conditions during transport) and mineral surface area (i.e., potential for OM protection), whereas our results show comparable $U^{k'}_{37}$ for all fractions. Prior authors (Rein et al., 2005; Kienast et al., 2012) speculated sedimentary $C_{37}$ alkenones at this location are skewed towards El Niño events arguing coccolithophores preferentially grow in oligotrophic waters. In fact, while coccolithophores generally dominate the phytoplankton community in oligotrophic waters their absolute abundance is highest in high-nutrient periods/regimes (e.g., Flores and Sierro, 2007). Alkenone producers *E. huxleyi* and *G. oceanica* are generally linked to eutrophic waters and periods of maximum primary productivity (Tyrrell and Merico, 2004). Recent work reveals a significant positive correlation between $C_{37}$ alkenone concentrations from a global surface sediment compilation and maxima





Chl*a* in overlying waters (Raja and Rosell-Melé, 2021). Accordingly, we suggest preferential alkenone production during the austral summer (Prahl et al., 2010), when surface waters are warmest and primary productivity is at its highest, is the most

feasible explanation for the warm bias from sedimentary alkenones observed at this location.

Alkenone ages from grain-size fractions at NAM are more dissimilar than at PER, and are 2000-3000 $^{14}$C yr older than coeval foraminifera. Moreover, alkenone-derived SST values among grain size fractions range from 15.8 °C to 17.9 °C (Figs. 4 and 6). Both sites are characterized by high productivity, low oxygen exposure and local deposition, and defined as "initial" depositional systems (Ausín et al., 2021) in terms of OC dispersal and deposition. Site-specific characteristics, such

as lower primary productivity and a broader shelf might favor a larger impact of hydrodynamic processes on sedimentary OC and alkenone signals at NAM. Our results suggest lateral supply of pre-aged alkenones influenced by hydrodynamic particle sorting and potentially originating from different locations on the margin, and are consistent with prior models of sediment transport by bottom and intermediate nepheloid layers leading to the formation of an upper slope OC depocenter (Inthorn et al., 2006a; Inthorn et al., 2006b). However, bulk SST only differs 0.8°C from annual averaged SST, indicating

that the apparent influence of hydrodynamic mineral sorting on sedimentary alkenone $^{14}$C age and abundance might not necessarily impart an equivalent bias in alkenone temperature signals. Past changes in the temperature gradient between the sites of alkenone production and deposition may, however, lead to larger and unnoticed SST biases in the sedimentary record.

Large alkenone age and temperature discrepancies among grain-size fractions are observed in NAT (Fig. 4). NAT is located

within the New England Mud Patch, where large amounts of FS advected by strong bottom currents and storm-induced transport of sand occurs (Goff et al., 2019), enhancing the oxygen exposure time of OM associated with both grain size fractions during transport. This mechanism would foster alkenone aging/input of pre-aged alkenones as well as selective degradation of C$_{37:3}$ in low-surface-area minerals, as observed for sand fractions. Sedimentary alkenones reflect a warmer signal than that of the overlying surface water, in contrast with the colder bias observed from offshore, slope sediments

(Hwang et al., 2014) explained by lateral advection of resuspended sediments from a colder upstream location (Hwang et al., 2009; Hwang et al., 2021). On the shelf (< 150 mwd), however, accumulation of advected fine silt sediments occurs under a west-directed transport, as shown by seismic profiles and the presence of active southwestward megaripples (Twichell et al., 1981; Goff et al., 2019). Lateral transport of fine sediments to this site from the Georges Bank to the east, as hypothesized by Twichell et al. (1981), would result in the entrainment and deposition of sedimentary alkenones carrying a warmer signal.

Large alkenone age offsets are also apparent among grain-size fractions from SMB. In this basin, the impact of hydrodynamic processes is strongly modulated by basin topography and by local variability in bottom water oxygen content, which can lead to differences in alkenone ages of flank and depocenter sediments (Mollenhauer and Eglinton, 2007). The fidelity of the sediment-SST signature in SMB may be coincidental, as the presence of aged alkenones in all size fractions indicates addition of allochthonous (advected) material. Indeed, bomb radiocarbon was present in co-eval planktic

foraminifera, whereas this was not detectable in corresponding alkenone samples. These results may imply rapid degradation of fresh alkenones and/or alkenone input from distal locations.



## Conclusions

Alkenone concentration, [14]C age and SST was determined in surficial sediments and corresponding grain-size fractions (clay, fine and coarse silt, and sand) retrieved from 6 continental margin settings.

Our results provide clear evidence for alkenone transport as a consequence of their intimate association with surfaces of fine-grained minerals; subsequent hydrodynamic mineral sorting and associated exposure to oxic degradation during transport imparts a strong influence on sedimentary alkenone signals. Alkenones preferentially reside within the fine silt fraction (2-10

μm) of sediments. Overall, this fraction is the largest alkenone contributor to marine sedimentary signals and exerts a predominant control on the alkenone concentration, [14]C age and derived SST values manifested in bulk sediments. Alkenone [14]C ages from FS (but also CS) indicate resuspension and protracted transport of alkenones from distant regions (or past time periods), suggesting that the intimate association of alkenones with fine grained sediments has important implications for the paleoreconstruction of primary productivity and SST based on alkenone concentrations and distributions.

Significant $U^{k'}_{37}$-SST variability is observed among grain size fractions. We suggest that the predominantly warmer-than-instrumental SST may reflect two alternative processes: 1) selective degradation of the tri-unsaturated $C_{37}$ alkenone attributed due to lower OM protection offered by larger particles under oxic conditions, and 2) systematic input of allochthonous alkenones synthesized in warmer waters. Further work is needed to determine the validity and importance of these scenarios.

Assessment of alkenone amount, [14]C age, and SST in grain-size fractions sheds important new light on processes controlling alkenone signatures in bulk sediments from the studied sites, including vertical settling of fresh material, lateral transport of allochthonous and pre-aged alkenones and alkenone degradation. The combined influence of alkenone-mineral associations and hydrodynamic particle sorting processes on sedimentary alkenone signals is discernable at all sites, ranging from almost negligible (e.g., at PER) to substantial (e.g., BER). Yet, pronounced impacts on alkenone [14]C age and concentration do not

necessarily impart an equivalent bias in $U^{k'}_{37}$-SST (e.g., SBM and NAM), as the latter also depends on the temperature gradient between the sites (or time periods) of alkenone production and deposition. Past changes in this temperature gradient could, however, lead to larger SST biases in the sedimentary record.

Our results highlight the importance of considering the influence of hydrodynamic processes (e.g., lateral transport) on sedimentary alkenone signatures (amount, age, and temperature) and their relationship to surface waters overlying the

depositional location.

**Data availability.** All original data used in this study, necessary to understand, evaluate, and replicate this research, are presented and available in tables within the main text.



**Author contribution.** B.A and T.I.E. conceived and designed this investigation. N.H. assisted with radiocarbon analyses.
E.B. assisted with grain-size fractionation. B.A. prepared and processed the samples, analyzed the results, and wrote the
manuscript with contributions by all coauthors.

**Competing interests.** The authors declare that they have no conflict of interest.

**Acknowledgements**
This study was supported by the project "TRAMPOLINE" (200021_175823) funded by the Swiss National Science
Foundation. The majority of the samples included in this study were collected as part of US National Science Foundation-
funded projects led by T.I. Eglinton and colleagues. We acknowledge the Regional Graduate Network for Oceanography
(RGNO) Discovery Camps, supported by the Agouron Institute, the Simons Foundation, the Scientific Committee for
Oceanographic Research (SCOR), the Ministry of Fisheries and marine Resources (MFMR), the National Marine
Information and Research Center (Nat MIRC), the University of Namibia (UNAM), ETH Zurich and the swiss i-research &
training institute, as well as scientists and crew of the R/V *Mirabilis* for realization of Namibian margin sampling. We are
grateful to all crew members involved in sample collection, in particular to Daniel Montluçon.

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
