# Peer review of "The influence of lateral transport on sedimentary alkenone paleoproxy signals"

_Biogeosciences, 2021_

## Author Comment (AC1)

We would like to thank the Editor and both reviewers for the time devoted on our manuscript and their valuable comments. We provide a point-by-point reply below or next to each comment in green.

Anonymous Referee #1

Specific comments:

Line 116: Table 1. the cores are in very different water depths, can you acknowledge that somewhere in the discussion, especially knowing that many deep-time paleotemperature and -productivity records come from open ocean deep-sea sediments. Added in line 399-400. Line 127: for the quantification, what standard and at what concentration was used? Did you run replicates? Concentrations were quantified using *n*-hexatriacontane as external standard.

Analytical precision of Uk'37 was better than 0.003 units based on repeated measurements of an inhouse alkenone standard. Both statements were added to the text. We also frequently checked the linearity of the GC-FID by injecting different volumes of the external standard, whose concentration was 12.49 ng/ul, but we do not think this data needs to be reported.

Line 129: Mention the exact variables that were used for the Bayspline code (e.g. prior standard deviation). Also state measurement/calibration error. Is it possible to show this

error in figure 5? In this version, we have used the calibration by Prahl et al. [1988], whose slope is the same used by BAYSPLINE for SST below 24 °C. Thus, the propagated error of SST estimates is 0.51 °C and it considers both, the analytical precision of SST and the  $1\sigma$  uncertainty of the calibration (0.5°C) reported by Prahl et al. [1988]. These errors are now plotted in Fig. 6A. In figure 5B, errors of SST offsets are propagated considering the error of SST estimates, and the propagated error of the annual SST, which considers the error given by ODV along latitude and along longitude.

Line 152: Even though this study seems to have better recoveries than other previous studies using wet-sieves (85%, Magill et al., 2018, Tesi et al., 2016), how can you be sure that there want he approximately are C27/2, which exclude ultimately affect.

that there wont be any preferential loss of C37:3 or C37:2, which could ultimately affect

your SST results? Could you comment on that? Preferential solubilization of labile OM within MilliQ water resulting in preferential loss of C37:3 or C37:2 during sample processing cannot be entirely excluded, but we carefully prepared our samples in the minimum possible amount of time (a matter of < 6 hours) to minimize OM exposure to water and OM loss during the fractionation process. The obtained fractions (along with the water in which some of them were collected) were immediately frozen and stored until they were freeze-dried. Considering the time identified in incubation studies for complete alkenone degradation under oxic conditions (2-3 months) [Sun et al., 2004], preferential degradation of the di- or tri-unsaturated alkenone due to free radical oxidation and aerobic bacterial processes [*Rontani et al.*, 2013] in MiliQ water in few hours cannot be entirely excluded but seems unlikely.

Line 153: Can you explain the significant loss of alkenones during manual columns? I lost part of the sample (1-2 drops) when loading it to the column with the pipette. I added a short sentence to the text. Instead of comparing the fraction-weighted values to the bulk values did you also use an

internal standard to account for the loss/recovery of alkenones? No internal standards were added to avoid the addition of extraneous C (having a different 14C signal) that would contaminate our samples, as these were intended for subsequent 14C dating.

Line 238: Can you state if you recorded C37:4 in BER, which also reflects advection of

high-latitude alkenones (Rosell-Melé et al., 1998; Bendle and Rosell-Melé, 2004; Bendle et

al., 2005)? There is a peak in the bulk sample of BER that is consistent with the expected retention time of C37:4. However, our identification alkenone standard only contains C37:3 and C37:2 and we cannot conclude whether the mentioned peak corresponds or not to the tetra-unsaturated alkenone without a maspec.

**Minor comments/ technical corrections:**

Line 16: spell out OM and SST as first mention in manuscript. Done.

Line 28:(C37:2 + C37:2) should be (C37:2 + C37:3). Corrected.

Line 33: also add latitude-dependent (Lutz et al., 2007). This work explores the vertical flux of POC and its results might not necessarily apply to alkenone fluxes.

Line 42: define 37:2, 37:3, e.g. UK37'=[C37:2]/([C37:2]+[C37:3]), where [C37:2] and

[C37:3] are the concentrations of di- and triunsaturated C37 alkenones, respectively. Done.

Line 44: reported precision in Prahl and Wakeham, 1987 is ±0.5°C. Done.

Line 55: wrong format for reference. Corrected.

Line 65: which hydrodynamic processes are we talking about? Maybe give an example in brackets? Done.

Line 68: define OC. Done.

Line 95-103: for consistency, add oxic/suboxic conditions to NAT, NAM and BER in text. Done. Line 109: Figure 1: Out of curiosity: why was SHAK06 not described in this study? It was analyzed in Ausin et al., 2021 for OC in bulk sediment on continental margin sediments. Because the remaining material from each fraction was not enough for alkenone quantification.

Line 114: add abbreviation to clay, similar to fine silt and coarse silt, which is later used in figures and tables. Done.

Line 115: The author refers in Ausín et al., 2021 to Magill et al., 2018 for the method. Might be worth mentioning here as well. Done.

Line 168: Figure caption 2: Figure 2. C) should be C37:3 and D) should be C37:2. Corrected. Define abbreviations for CS, FS, C. Done.

Line 171: briefly mention, why alkenone 14C ages in SBB, NAF, BER were not measured. Done. Line 185: Figure 3 would be easier to read if you would change TOC-foraminifera to another symbol. It currently might be confusing because you use open symbols in panel A for bomb 14C. Done.

Line 190: Figure 4: Define abbreviations for CS, FS, C. Done (also in Figs. 2 and 6). Line 192: Fig.5. to be consistent with the naming in figure 6C, rename y-axis in panel B " $\Delta$ SSTbulk-annual mean". Done.

Line 275: Figure 7: In panel A, x-axis label is missing. Added. To be consistent with the nomenclature of your manuscript rename label for y-axis to OC%. Done. Also, insert is hard to read, maybe its enough to state the R2 from Ausín et al., 2021 in the text, instead of having the insert. Changed as suggested. In panel B, add C37 to y-axis to be consistent with naming (in comparison to Fig. 2A). Done.

**References**

Prahl, F. G., L. A. Muehlhausen, and D. L. Zahnle (1988), Further evaluation of long-chain alkenones as indicators of paleoceanographic conditions, *Geochimica et Cosmochimica Acta*, *52*(9), 2303-2310. Rontani, J. F., J. K. Volkman, F. G. Prahl, and S. G. Wakeham (2013), Biotic and abiotic degradation of alkenones and implications for U37K' paleoproxy applications: A review, *Organic Geochemistry*, *59*, 95-113.

Sun, M.-Y., et al. (2004). "Molecular carbon isotopic fractionation of algal lipids during decomposition in natural oxic and anoxic seawaters." Organic Geochemistry **35**(8): 895-908.

---

## Author Comment (AC2)

We would like to thank the Editor and both reviewers for the time devoted on our manuscript and their valuable comments. We provide a point-by-point reply below or next to each comment in green.

**Anonymous Referee #2**

Major comments:
Concept vs new findings: The concept of the influence of lateral transport on the age and SST estimates of alkenones is more than 20 years old (see Benthien & Müller, 2000 and diverse follow-up papers). Now this manuscript presents grain-size specific SST and 14C data which support this concept and point to the fine silt fraction as being mainly responsible for the observed effects. My main concern with the Ausín et al. paper is the way this is presented. The history of this concept is described in the introduction but it nevertheless reads as if the whole concept is new and Ausín et al. are indeed the first to present the concept. I am referring to sentences like: "Our results demonstrate that selective association of alkenones with mineral surfaces and associated hydrodynamic mineral sorting processes can alter alkenone signals encoded in marine sediments (14C age, content, and distribution) and confound corresponding proxy records (productivity and SST) in the spatial and temporal domain.". In this last sentence of the abstract, the first half of the sentence reads as if this was not known before. In contrast, this has been clear before and is just supported by the new data. The second half of the sentence, in contrast, is just an inference so far. It is logically to expect these effects but it is not actually demonstrated in the manuscript. Another example is the last sentence of the conclusions: "Our results highlight the importance of considering the influence of hydrodynamic processes (e.g., lateral transport) on sedimentary alkenone signatures (amount, age, and temperature) and their relationship to surface waters overlying the depositional location.". Since the works from Mollenhauer, Ohkouchi and others from decades ago it is known that the influence of hydrodynamic sorting on organic paleo signals in sediments has to be considered. Such sentences thus read like 'constructing a strawman' to oversell own findings. That said, I think the new grain-size specific data in itself carry enough value to be reported and the manuscript does not benefit from overselling and should be adjusted to be toned down.

The influence of lateral transport on alkenone 14C ages has been largely explored in detail by other authors since the pioneering work by Ohkouchi et al. [2002] almost two decades ago. In fact, the work by Benthien & Müller, 2000 was also cited. This is explained in the introduction (lines 43-46 and lines 78-81) and we do not intend to present such knowledge as a new concept arising from our results. We have added another paragraph to provide more background information (lines 46-49). By contrast, our study explores new research questions that, to our knowledge, have not been addressed yet: For instance, what is the role of alkenone-mineral associations regarding their preservation/degradation? To date, this question has been addressed for organic carbon and other specific biomarkers like GDGTs and fatty acids [e.g., *Peterse and Eglinton*, 2017] but not for alkenones. Also, are there specific mineral grain-size fractions that play a major role in the advection of allochthonous alkenones, or do they all have the same potential to introduce allochthous alkenone signals in marine sediments? Given the lack of other studies exploring sedimentary alkenone signals in fractionated sediments, these question has remained elusive, and we believe our results are novel in that regard. Finally, what can be said about the impact of allocthonous alkenones on alkenone-proxy signals? The vast majority the studies exploring lateral transport of sedimentary alkenones emphasize that such mechanism may cause temporal offsets giving rise to biases in the recorded proxy signal, but there have been very few attempts to estimate impacts on corresponding proxy-signals (SST through Uk'37 ratios and productivity through alkenone abundance). In sum, the novelty of our study resides in: i) the assessment of the role of alkenone-mineral relationships, a dimension that is missing in other works; ii) the assessment of the impact of hydrodynamic mineral sorting (specific to size fractions) on sedimentary alkenone signals; and iii) how i and ii impact corresponding alkenone proxy signals preserved in marine sediments.
In any case, the specific sentences mentioned by the reviewer have been rephrased to emphasize which concepts are known since long ago.

Biological oceanography: Another point which seems to be wrong is the interpretation of

alkenones being indicative for highest productivity in the Peruvian upwelling system used for explaining a warm bias (line 323 to line 325). This cannot be true as upwelling activity is driven by trade wind strength which is highest in austral winter leading to deep Ekman pumping which brings dissolved Si into the surface waters causing outcompeting of haptophytes by diatoms. The warm bias thus likely arises from the fact that alkenones are actually not produced during strongest upwelling, the latter associated with lowest SST.

In the Peruvian upwelling system, upwelling activity is highest in austral winter, but surface chlorophyll concentration is highest in austral summer and decreases during austral winter, in phase opposition with coastal upwelling intensity [e.g., *Echevin et al.*, 2008]. The paradoxical seasonal cycle of this region has been studied by other authors and is out of the scope of this paper. Regarding the blooming season of coccolithophores, a general misunderstanding exists. The latter mostly derives from the fact that coccolithophores generally dominate the phytoplankton community in oligotrophic waters (they outcompete diatoms in these cases). However, this fact does not imply that their absolute abundance is higher there than in productive waters. In fact, they appear in higher numbers when Chla concentrations are higher [e.g., Flores and Sierro, 2007]. This is, they might not dominate the assemblage in productive waters, but they are more numerous than in fully oligotrophic waters/periods. Indeed, diatoms outcompete coccolithophores in more productive waters and dominate the phytoplankton community at the peak of the upwelling (highest nutrient concentration, higher turbulence and colder waters). Yet, as shown in studies of spatial and temporal variability of coccolithophore productivity in upwelling regions, coccolithophores bloom right after diatoms, showing a strong preference for more mature upwelled waters [*Ausín et al.*, 2018; *Mitchell-Innes and Winter*, 1987; *Silva et al.*, 2008]. This is, more stable, warmer and lower nutrient waters, but still during the more productive season. In the sediments, this fine-scale temporal evolution (matter of days) is lost, and higher number of coccoliths are widely used in as an indication of higher net primary productivity. Specifically for the coccolithophore species that are responsible for alkenone production, and as stated in the text: "Alkenone producers E. huxleyi and G. oceanica are generally linked to eutrophic waters and periods of maximum primary productivity (Tyrrell and Merico, 2004). Recent work reveals a significant positive correlation between C37 alkenone concentrations from a global surface sediment compilation and maxima Chla in overlying waters (Raja and Rosell-Melé, 2021)". Therefore, we believe our reasoning remains a feasible scenario to explain our results.

Errors and precision: Please state what the analytical and propagated errors are of compound quantifications and SST estimates. For instance, in table 2 alkenone concentrations (C37:2 and C37:3 and combined) are reported to the last digit. Is this reasonable with a usual GC-FID error of at least 10% for long-chain alkenone quantification? How does this error propagate to the UK37'-SST estimates? I guess that with a good error handling many of the reported 'biases' will actually be within error and only a few significant offsets will remain. Also, error bars should be added to all plots showing SST estimates and, preferentially, also instrumental SST data. We would like to thank Reviewer 2 for this critic and relevant comment on our paper. We agree that the paper could be improved in this regard and have therefore modify it accordingly. Analytical precision of Uk'37 was better than 0.003 units based on repeated measurements of an in-house alkenone standard. This statement has been added to the text. Regarding SST errors, we have added corresponding $1\sigma$ uncertainties of SST estimates derived from error propagation considering the analytical precision of SST and the $1\sigma$ uncertainty of the calibration (0.5°C) reported by Prahl et al. [1988]. These errors are now plotted in Fig. 6 and mentioned in the header of Table 2. Errors for age offsets have been also propagated, and consider both, the propagated error of SST estimates and that of the annual-mean atlas SST. When these errors are considered, warmer SST offsets in relation to instrumental values for PER and NAT (and NAM to lesser extent) remain. We have modified the text accordingly and deleted the related paragraph from the conclusions as a "general warm bias" is not observed when considering errors.

Minor comments:
Generally, please replace 'warmer bias' by 'warm bias'. Done.
Line 3: Please check affiliations. Bruni and Eglinton are not in Salamanca. Done.
Line 9: "…gaps remain on…" - consider wording. Done.
Line 29: Are alkenones really a large component of total OC of Emiliania huxleyi? Please check, I doubt this statement. We paraphrased and cited Prahl et al. [1988]: "*The long-chain alkenones constitute a major component (8.0 ± 2.9%) of the total organic carbon content of living*

*cells of E. huxleyi*". We did not find other estimates elsewhere, but considering the wide variety of organic molecules that contribute to TOC, we agree with Prahl et al. that 8.0 ± 2.9% can be considered a major contribution.

Line 106: "…in contrast with…" – consider wording 'contrast to'. Done.

Line 113: how was the grain-size fractionation done? Wet or dry sieving? Added.

Line 201: I would presume that the statement that only SST estimates from SBB and SMB are comparable to atlas data is not true when considering errors. Avoid arguing with 'comparability' and argue with errors instead. When considering errors, the following applies: "Sediment and atlas SST values from SBB, SMB and NAF fall within the associated uncertainties whereas temperature differences ranging from -6±0.6°C to +3±1.1°C are observed at PER, NAT, NAM and BER (Fig. 5B)" Also in the discussion "Alkenones from SMB, SBB and NAF are found to reflect local instrumental SST within associated errors (Fig. 5), while a positive discrepancy ranging from 0.8±0.5°C to 3±1.1 °C (towards warmer temperatures) is observed at other locations with the exception of BER (-6 °C±0.6°C)"

Line 204: I doubt that the statement of a general warm bias is actually true when considering uncertainties. Looks like only true for 2 out of 7 samples. Rephrased to: "SST discrepancies imply core-top SST is significantly warmer than surface water temperature at PER and NAT". NAM and NAF also show a warmer bias within the error, but much smaller.

Line 208: The statement that FS overall shows the smallest temperature offsets with bulk is not true. Please look at the data from NAT. I do not understand the significance of the following statement on larger/smaller offsets. Consider removing. We have rephrased it and replaced "overall" by "Except for PER and NAT" to be more precise. At SBB, SMB, NAM, BER and NAF, fine silt is the fraction that shows a SST most similar to that of bulk, and this is important because it highlights this fraction largely contributes to the SST signal measured in bulk sediments (via its high sediment mass contribution and alkenone content). Considering FS is prone to resuspension and might have been transported from other regions, it might be introducing a bias in $SST_{bulk}$. In fact, alkenone 14C results from FS samples indicate they contain pre-aged or older alkenones, already suggesting the asynchronous (thus, possibly allochthonous) origin of these biomarkers.

Line 318-321: In upwelling areas coccolithophores are outcompeted by diatoms during strong upwelling. See major comment above. Please see our reply above.

Line 380: should read SMB. Done

Tables:

Table 1: Namibian core – MC or BC? It is a MC, but the original name of the sample does not include this information in it labelling code. We prefer to keep the original naming.

Table 2: see comment on precision of data given. Please see comment above.

Figures:

Figure 2: Please add comment in caption about sand fraction in BER. Done.

Figure 4: I do not see the significance of panel B, just another representation of the same data (also no reference to B in caption). Panel A are the age results, panel B are the derived age offsets. Both panels are typically presented in works of this nature (please see Mollenhauer et al. [2005]) to help age offset visualization. Reference to B has been added to the caption.

Figure 5: panel B: see comment on errors. I suspect that data from SBB, SMB, NAM and NAF are actually within error with instrumental SST. Preferentially add error bars to SST estimates and instrumental SST data. In figure 5B, errors of SST offsets between measured and annual means are propagated considering the error of SST estimates, and the propagated error of the annual SSTs obtained from Ocean Data View (ODV), which considers the error given by ODV along latitude and along longitude. The propagated error of annual mean SSTs is given in Table 1.

Figure 6: captions for panel B and C are mixed up. Corrected.

Figure 8: What is the significance of the solid regression line considering all data. Consider removing. Removed.

**References**

Ausín, B., et al. (2018), Spatial and temporal variability in coccolithophore abundance and distribution in the NW Iberian coastal upwelling system, *Biogeosciences*, *15*(1), 245-262.

Echevin, V., O. Aumont, J. Ledesma, and G. Flores (2008), The seasonal cycle of surface chlorophyll in the Peruvian upwelling system: A modelling study, *Progress in Oceanography*, *79*(2), 167-176.

Mitchell-Innes, B. A., and A. Winter (1987), Coccolithophores: a major phytoplankton component in mature upwelled waters off the Cape Peninsula, South Africa in March, 1983, *Marine Biology*, *95*(1), 25-30.

Mollenhauer, G., M. Kienast, F. Lamy, H. Meggers, R. R. Schneider, J. M. Hayes, and T. I. C. P. A. Eglinton (2005), An evaluation of 14C age relationships between co-occurring foraminifera, alkenones, and total organic carbon in continental margin sediments, *Paleoceanography*(1), PA1016.

Ohkouchi, N., T. I. Eglinton, L. D. Keigwin, and J. M. Hayes (2002), Spatial and Temporal Offsets Between Proxy Records in a Sediment Drift, *Science*, *298*(5596), 1224-1227.

Peterse, F., and T. I. Eglinton (2017), Grain Size Associations of Branched Tetraether Lipids in Soils and Riverbank Sediments: Influence of Hydrodynamic Sorting Processes, *Frontiers in Earth Science*, *5*(49).

Prahl, F. G., L. A. Muehlhausen, and D. L. Zahnle (1988), Further evaluation of long-chain alkenones as indicators of paleoceanographic conditions, *Geochimica et Cosmochimica Acta*, *52*(9), 2303-2310.

Silva, A., S. Palma, and M. T. Moita (2008), Coccolithophores in the upwelling waters of Portugal: Four years of weekly distribution in Lisbon bay, *Continental Shelf Research*, *28*(18), 2601-2613.

---

## Author Response (AR1)

**Rebuttal letter**

On behalf of all coauthors I would like to thank the Editor the time devoted to read and comment on our Manuscript and their suggestions to improve it. Below is a point-by-point reply in green, explaining the changes that we have made accordingly.

With all the abbreviations I found it a bit difficult to keep track of everything, if you see a way of improving this that would be highly appreciated. We have substituted FS by "fine silt", CS by "coarse silt" and C by "clay" in text, figures and tables. We would like to keep the abbreviation for the sample sites so they are comparable to a previous published work (Ausín et al., 2021) based on these very same samples and related size fractions, but we have written the extended name whenever possible.

This is also a little bit due to the tables format, which is partially a problem because of the portrait layout. One of the things I directly started looking for was a comparison between the results of the bulk sediment and a weighted average of the different sediments fractions and it took me a while to find this. I feel this could be emphasized a bit better, possible by adding the weighted average to the graph as well as the table? We have added this data to figures 2 and 6. I also think this should hold true not only for the amounts of alkenones, but also the ages, right? It should, but we would need the age of all the fractions contributing to the bulk to estimate the abundance weighted age, otherwise the result is not comparable. Unfortunately, purification of alkenones for radiocarbon dating was only possible in some of the size fractions, but not in all of them in any of the sites.  I think there is one sample where the alkenones from the bulk sediment were much younger than the alkenones from any of the sediment fractions? I could be wrong. In SMB and NAR the bulk is much younger than the related fractions, but it should be kept in mind that there are two other fractions for each site that we could not measure (clay and fine silt in SMB and clay and coarse silt in NAT). According to the 14C age of the OC hosted by each fraction (Ausin et al., 2021), clay is always the youngest fraction. Given 14C age in alkenones and OC from the same sample is comparable (Fig. 8), it is expected that 14C age of alkenones hosted by the clay fraction are also younger, shifting the average weighted ages to younger ages, and likely leading to fraction-weighted average ages comparable to bulk ages, as a small contribution of young material (high 14C content) has a large impact on samples containing little 14C.  For the 14C data I got a bit confused by "bulk" in table 3, I think, this is still the alkenones extracted from the bulk sediment, so completely different from TOC. I think you were correct, but if you can make that more instantly clear to the reader, I think that would help. The "bulk"-weighted average comparison is a really good indicator for the quality of the results. Indeed, these are the radiocarbon results of the alkenones only. We have stated this more clearly in the Table header.

I don't think you should write an extensive M&M section, however a few more details would be nice. Alkenones were identified by GC based on comparison with a standard, it would be nice to have some idea of the GC column type and length that was used. The samples were quite marine so I don't expect alkenones derived from freshwater (Type I) haptophytes, but they make an C37:3 (and other :3s) different from the "typical" marine one. On a typical apolar GC column these C37:3s cannot be separated from each other, for instance. Not a

problem for these setting, I would think. Multiple alkenone producing haptophytes do not produce coccoliths, be careful with the comparison between the two. We used 2x60m VF-5MS columns (120 m total) that allow good separation of C37 alkenones. We have added some information on the characteristics of the column and the standards used.

You indicate additional cleanup of alkenones for 14C measurements, however, I did not see any indication on how pure they actually were, I think? We only measured those samples that had a purity >90%. This information has been added to the text, but on average, our samples had a purity >96%.

Some minor comments, I found the sentence starting in line 242 to 244 not clear. At BER, we obtained a cold bias of 6°C±0.6°C, but as we did not use surface or core-top (0-1 cm) sediments, but sub-surface sediments whose planktic foraminifera are 900 yr old, it could be argued that the large temperature bias is due to the comparison of SST during two different periods. For this reason, we need to emphasize our results can be still discussed, as the bias between annual-mean and core-top sediments from that exact core is still 6.6 °C based on previous studies. Our sentence now reads: The large and cold bias observed at Bermuda Rise, could be due to the use of sub-surface sediments (2-5 cm; foraminifera $^{14}$C age=900±50 yr) rather than surface sediments. Nevertheless, alkenone-SST from the core-top (0-1 cm) of the exact same core [*Nao Ohkouchi et al.*, 2002] also leads to a -6.6°C (cold) bias.

In line 283 "14C ages of both types of organic matter". Corrected.

"Previous authors" is a bit strange, "studies have shown", "previous publications indicate". Changed.

I also noticed that after agreeing with reviewer 2 on "warm bias" instead of warmer bias in your answer to the next question you used warmer bias again. Be careful with the revised manuscript. Bad habit, noted.

You mention in your response that there was only one reference with an indication of the amount of alkenones per biomass. In Chivall et al., 2014 (OG) there are alkenones in pg per cell for culture studies, for instance, but I am sure there are more papers that indicate amounts of alkenones per unit biomass. We meant estimates in that (*E. huxleyi*) or other coccolithophore species that produce alkenones. Also, we meant the relative contribution of alkenones to the total organic carbon, which is the data we cite from Prahl et al. (1996). But we agree with the Editor that if this data is reported as pg of alkenones per cell of *E. huxleyi* (1.20±0.28 pg/cell; Prahl et al. 1996) comparable estimates can be found elsewhere (e.g., Sawada et al., 1996, Marlowe, 1984; Prahl et al., 2003).

If I look at figure 5, I seem to see two outliers, BER and NAT, if you remove these your correlation will be much better than 0.26. There is one other sampling site that does fit with the 1:1 line more or less, but all the alkenones are already a couple of thousand years old, if I read the manuscript correctly. They fit accidentally or not that much has changed in this time period. Or, they were synthesized in a place — because a temporal bias might also imply a spatial bias — with a similar SST, this is, little or no SST gradient between place of origin and final deposition. This line of argument is discussed in throughout the text and explicitly mentioned in the conclusions.

I agree with your warning to all of us to be careful, however I think you could end more positively by including recommendations and not only warnings. Now that we know this how can we deal with it? Multiproxy approach, combining foram and organic proxies, etc. I am looking forward to your rebuttal and the revised version of your manuscript based on the reviewers and my comments. We have shaped our conclusions accordingly. We believe further steps include the assessment of particle provenance to evaluate particle transport pathways and whether the entrainment of allochthonous/asynchronous material has or not a large impact on proxy signals. This is because the large impacts observed on alkenone 14C age and concentration do not necessarily impart an equivalent bias in Uk'37-SST (as observed at sites SBM SMB and NAM), because the latter also depends on the temperature gradient between the sites (or time periods) of alkenone production and deposition. If this gradient is small or even negligible, it is good news to the Paleoceanographic community. We also agree multiproxy approaches are still needed to evaluate the accuracy of proxy signals and the reliability of derived climate interpretations, specially those combining organic and inorganic proxies and based on proxy carriers that show different hydrodynamic behavior.

References

Ausín, B., E. Bruni, N. Haghipour, C. Welte, S. M. Bernasconi, and T. I. Eglinton (2021), Controls on the abundance, provenance and age of organic carbon buried in continental margin sediments, Earth and Planetary Science Letters, 558, 116759.

Marlowei. T., Greenj . C., Neala . C., Brassells. C., Eglinton G. and Course P. A. (1984) Long chain alkenones in the Prymnesiophyceae. Distribution of alkenones and other lipids and their taxonomic significance. Br. phycol. I. 19,203-2 16.

Prahl, F. G., Wolfe, G. V., and Sparrow, M. A. (2003), Physiological impacts on alkenone paleothermometry, Paleoceanography, 18, 1025, doi:10.1029/2002PA000803, 2.

Sawada, K., Nobuhiko Handa, Yoshihiro Shiraiwa, Akiko Danbara, Shigeru Montani. (1996) Long-chain alkenones and alkyl alkenoates in the coastal and pelagic sediments of the northwest North Pacific, with special reference to the reconstruction of Emiliania huxleyi and Gephyrocapsa oceanica ratios, Organic Geochemistry, 24, 751-764.

---

## Author Response (AR2)

Dear Editor,

We very much appreciate the time you devoted to reviewing our revised Manuscript and for providing constructive comments that improved its quality.
We have edited the text according to all the suggestions, including the use of acronyms. A few clarifications:

*Line 39: "….. transport which is…."* The sentence was correct, but too long, we have shortened.

*Line 165/166: total lipid loss? As in the extract never reached the column?* Indeed, I lost a drop of concentrated total lipid extract while transferring it from the vial to the column.

We have adjusted Table 2 and it fits and reads better now.

Regarding figure 5, we haven't found a good way to modify it as to make it more self-explanatory. We could mark as "old" the alkenones for which we have data for both, forams and alkenones (i.e., NAM, SBB, SMB), but none of these three look like outliers. For the other samples, rather we miss the 14C of forams as time reference, or we miss the 14C age of the alkenones. We know, nevertheless, these alkenones are old based on the strong positive correlation existing between 14C age of OC and co-deposited alkenones (as shown in fig. 8), and because of previous work from the same core (Ohkouchi et al. 2002). However, to explain this in the figure caption we would need to recall to relationships that are explained later in the text, and we believe we add more confusion this way.

I wish you happy holidays and new year on behalf of all the authors.

Blanca Ausín and Co-authors.